

# Improved calibration of Green-Ampt infiltration in the EROSION-2D/3D model using a rainfall-runoff experiment database

Hana Beitlerová[1], Jonas Lenz[2], Jan Devátý[3], Martin Mistr[1], Jiří Kapička[1], Arno Buchholz[2], Ilona Gerndtová[4], and Anne Routschek[2]

[1]Research Institute for Soil and Water Conservation, Prague, Czech Republic
[2]Soil and Water Conservation Unit, TU Bergakademie Freiberg, Freiberg, Germany
[3]Czech Technical University in Prague, Prague, Czech Republic
[4]Research Institute of Agricultural Engineering, Prague, Czech Republic

**Correspondence:** Hana Beitlerová (beitlerova.hana@seznam.cz)

**Abstract.** Soil infiltration is one of the key factors that has an influence on soil erosion caused by rainfall. Therefore, a well-represented infiltration process is a necessary precondition for successful soil erosion modelling. Complex natural conditions do not allow the full mathematical description of the infiltration process and additional calibration parameters are required. The Green-Ampt based infiltration module in the EROSION-2D/3D model is adjusted by calibration of the skinfactor parameter.

Previous studies provide skinfactor values for several combinations of soil and vegetation conditions. However, their accuracies are questionable and estimating the skinfactors for other than the measured conditions yields significant uncertainties in the model results. This study presents new empirically based transfer functions for skinfactor estimation that significantly improve the accuracy of the infiltration module and thus the overall EROSION-2D/3D model performance. The transfer functions are based on a statistical analysis of the rainfall-runoff simulation database, which contains 273 experiments compiled by two

independent working groups. Linear mixed effects models, with a manual backward elimination approach for the predictor selection, were applied to derive the transfer functions. Soil moisture and bulk density were identified as the most significant predictors explaining 79% of the skinfactor variability, followed by the soil texture and the impact of previous rainfall events. The mean absolute percentage error of the skinfactor prediction was improved from 192% using the currently available method, to 66% using the presented transfer functions. Error propagation of the predicted skinfactors into the surface runoff and soil

loss on the hypothetical slope showed significant improvement in the EROSION-2D/3D results. A first validation of real rainfall-runoff events indicates good model performance for events with a higher total precipitation and intensity.

## 1 Introduction

Soil erosion modelling is a common and efficient approach to analyse and understand the soil erosion process and propose solutions to minimize its impact. Therefore, development and improvement of soil erosion modelling tools are of crucial

interest among soil scientists, state land offices, or landscape architects. EROSION-2D and EROSION-3D are soil erosion modelling tools based on the same physical descriptions of soil erosion processes on hillslopes (2D) or in catchment areas (3D)





for single rainfall events. In this paper EROSION-2D/3D shall refer to both versions, where shared algorithms are discussed. These tools are able to predict erosion patterns, as well as deposition areas, on agricultural fields, infrastructure, and settlement areas (von Werner, 2007). The physical based algorithms allow to apply EROSION-2D/3D under various circumstances, from long term simulations, covering catchments of several square kilometres (Routschek et al., 2014), to short term reconstructive

simulations of small catchments (Hänsel et al., 2019).

EROSION-2D/3D includes two submodules. The first submodule is an infiltration module used to calculate infiltration rates over time. The second submodule uses the infiltration rates to calculate excess water, surface runoff, and detachment, as well as the transport and deposition of particles. The infiltration submodule is based on the Green-Ampt approach (Schmidt, 1996). This approach assumes a rigid, homogenous, and permanent submerged soil column, which does not usually allow the

simulation of natural conditions without additional calibration parameters or advanced algorithms. The infiltration submodule in EROSION-2D/3D requires input parameters that can be measured or predicted with common methods (i.e., bulk density, initial soil moisture, grain size distribution, and organic bound carbon) and the skinfactor calibration parameter. The skinfactor can be determined from rainfall-runoff or infiltration experiments with the hillslope simulation tool EROSION-2D (Michael et al., 1996). This process requires extended time and demands manual labour, limiting the skinfactor determination to a

relatively small number of combinations of soil and vegetation conditions.

Previous studies have focused on estimating skinfactors for those other than measured conditions. The studies are based on 116 rainfall experiments conducted in Saxony (Germany) between 1992 and 1995, which are published in the EROSION-3D Catalogue of Input Parameters (Parameter Catalogue) (Michael et al., 1996). Michael et al. (1996) and von Werner (2009) estimated the skinfactors using information on German KA5 soil textural classes (Sponagel and Ad-hoc-Arbeitsgruppe Boden,

2005), initial soil saturation (dry or wet conditions), plant development stages, management practices, and field conditions. All of the predictors were factorial variables. The resulting matrix of skinfactor values provides guidance for a limited number of vegetation and soil condition combinations, which is available in the Parameter Catalogue for model users. However, the statistical background of the matrix and the selection of the predictors were not published and are not traceable. For other conditions, users must estimate values by themselves from the limited and incomplete matrix. Another approach (Michael,

2000; Schlegel, 2012) was to predict skinfactors from the numeric soil input parameters of the infiltration module (i.e., clay, silt, sand, organic carbon, bulk density, and soil moisture). Both studies used regression models to analyse the strongest predictors for different groups of experiments according to the soil types, management practices, and moisture conditions. The entire dataset shows the strongest correlation between the skinfactor and the bulk density, soil moisture, and silt content, but with a low statistical significance and small correlation coefficient. Analysis of specific groups of experiments (e.g.,

sandy soils and conservational management practices) exhibits better results, but are based on an insufficient number of experiments. For this study, an R package toolbox.e3d was developed to enable automatic and batch determination of the skinfactors for multiple rainfall-runoff infiltration experiments. An extensive rainfall-runoff experiment database was processed by the package, creating a sufficient amount of data to statistically analyse the relationships between the skinfactor and other parameters describing the soil and vegetation conditions of the experiments. The aim of this study is to improve the performance of EROSION-2D/3D by providing easy to use transfer functions to calibrate the infiltration module of the model. This paper



reports the skinfactor transfer functions derived from currently available data; however, this process is fully reproducible using the R code provided in the supplementary material of this paper, such that the functions can be improved and more robustly validated using the growing dataset of rainfall simulations.

## 5    2    Data and methods

### 2.1    Skinfactor

The infiltration submodule used in EROSION-2D/3D was developed by Schmidt (1996) based on the Green-Ampt infiltration approach (Green and Ampt, 1911), which includes a simplification of the infiltration process by assuming that rainwater penetrates the soil in a piston-like flow and completely saturates the available pore space. Empirical functions are used to 10    estimate the matrix potential (Vereecken et al., 1989; Van Genuchten, 1980) and saturated hydraulic conductivity (Campbell, 1985). A full description of the infiltration submodule is given in Schmidt (1996). As the theoretical concept of infiltration assumes a rigid soil matrix, time variable structural processes, such as soil compaction, slaking, and crusting or macropores due to shrinking and biological activities, should be considered using an empirical factor, known as the skinfactor. This factor is used to adjust the saturated hydraulic conductivity $k_s$ according to Schindewolf and Schmidt (2012) as follows:

15    $$k_s = k_{sat} \cdot skin \tag{1}$$

where $k_{sat}$ is saturated hydraulic conductivity $[\mathrm{kg\,m^{-3}\,s^{-1}}]$ as calculated by Campbell estimation, $k_s$ is saturated hydraulic conductivity adjusted by skinfactor $[\mathrm{kg\,m^{-3}\,s^{-1}}]$, and $skin$ is skinfactor $[-]$.

Values of the skinfactor <1 reduce the infiltration rate to consider the effects of soil slaking and crusting, as well as anthropogenic compaction. Values of the skinfactor >1 cause a positive correction of the infiltration rate, e.g., to consider 20    increased infiltration in macropores due to soil shrinking, biological activity, or tillage impact. Two methods of deriving the skinfactors from rainfall-runoff experiments were established in previous studies, both yielding slightly different values, resulting in different surface runoff rates. The first established method uses the skinfactor to adjust the amount of cumulative runoff from the plot area (skinfactor$_{\text{runoff}}$) (Michael, 2000). The second established method uses the skinfactor to adjust a certain infiltration rate, usually the final infiltration rate at the end of the experiment (skinfactor$_{\text{inf}}$) (Schindewolf and Schmidt, 25    2012). The best method remains a topic of debate among model developers. In this study, we used both methods to derive the skinfactors for the analysis. Transfer functions for the skinfactor$_{\text{inf}}$ showed a better fit to the validation datasets and are therefore presented in this study.

### 2.2    Rainfall-runoff data

An open database for storing, maintaining, and sharing protocols from rainfall-runoff experiments is being developed in parallel 30    to this study (Deváty et al., 2020). Currently, the database contains protocols from three working groups: The Technical University of Freiberg, Germany (TUBAF); the Research Institute for Soil and Water Conservation, Czech Republic (RISWC);



and the Czech Technical University in Prague, Czech Republic (CTU). The database contains 464 experiments (126 from TUBAF, including the original 116 experiments used in previous studies, 191 from RISWC, and 147 from CTU). Experiments contained in the database were conducted for different projects and purposes. Not all experiments contain all input parameters

required for skinfactor calibration, where the methodology of data acquisition and analysis can differ between working groups. The CTU data do not contain organic carbon content and bulk density and were thus not used in this study. Another 44 RISWC and TUBAF experiments were excluded from further analysis due to missing input parameters, no generated runoff, or the use of non-standard experimental conditions. Factorial predictors of crops and management practices were fractionated into many levels represented by a few to tens of cases. For better statistical representation, the predictors were categorized into subgroups

based on their similar behaviour during the erosion process (Table 1). The complete and consolidated dataset for statistical analysis contains 273 RISWC and TUBAF experiments. Parameters included in the statistical analysis and respective data acquisition methods used by the working groups are listed in Table 2.

**Table 1.** Reduction of factorial variables Crop and Management practice.

| crop levels grouping | |
| --- | --- |
| seedbed | seedbed |
| erosion permitting crop | maize, potatoes, root beet, sunflower |
| legume | broadbeen, peas, flax, lupine |
| oilseed crop | white mustard, oilseed rape |
| cereals | spring barley, winter barley, spring wheat, winter wheat, winter rye, panic grass |
| catch crop, erosion restricting crop | ryegrass, field pea, buckwheat, purple tansy |
| management levels grouping | |
| conventional tillage (CvT) | CvT, CvT with removed stones, CvT after grass, CvT with undersowing |
| conservational tillage | low tillage, chiselled, vertical tillage after field pea, vertical tillage after white mustard, vertical tillage after purple tansy |
| no tillage (NT) | NT to mulch, NT after desiccated white mustard, NT after desiccated ryegrass, NT after desiccated field pea, NT after desiccated purple tansy |

## 2.3 Skinfactor prediction

The skinfactor has a nearly logarithmic distribution, with values ranging from 0.001 to 100 in the dataset. The assumption

of normally distributed residuals in the linear mixed effects models used in this study is violated when using untransformed skinfactors. Logarithmic transformation of skinfactors produces a near normal distribution for the residuals. Therefore, this transformation was used for all skinfactor values in the statistical analysis.

The dependency of the skinfactor on single predictors was tested in the correlation matrix for the numerical predictors and via an ANOVA analysis for the factorial predictors to obtain the first insight into the relationships. Numerical variables of the initial soil moisture, bulk density, and soil texture were correlated with the skinfactor. Multicollinearity was observed between




**Table 2.** Parameters included in statistical analysis for skinfactor prediction.

| parameter | method TUBAF | method RISWC | type of variable | category/unit |
|---|---|---|---|---|
| skinfactor | EROSION-2D iterative determination | EROSION-2D iterative determination | float | - |
| clay/silt/sand content* | no standard used - dispersion methods H2O, chemicals, ultrasound | pipeting method | float | $M - \%$ |
| soil texture class* | soil texture triangle | soil texture triangle | factorial | clay/silt/loam/sand |
| organic carbon | combustion method of disturbed soil samples | Walkley-Black chromic acid wet oxidation method | float | $M - \%$ |
| bulk density | dried soil core cylinders | dried soil core cylinders | integer | $kg\,m^{-3}$ |
| initial soil moisture | TDR-probe in field, repeated gravimetrical measurement of soil core cylinder | TDR-probe in field, repeated gravimetrical measurement of soil core cylinder 5-10 cm depth | float | $V - \%$ |
| soil saturation | dry run - natural soil moisture conditions, wet run - after dry run reached steady infiltration and break up to one day | dry run same as FG, wet run - after 30 min dry run and 15 min break | factorial | dry /wet |
| crop | crop name | crop name | factorial | 6 categories (see Table 1) |
| vegetation cover | estimation in field | supervised picture classification | integer | % |
| time of consolidation | days from last topsoil disturbing operation | days from last topsoil disturbing operation | integer | days |
| management practice | manag. name | manag. name | factorial | 3 categories (see Table 1) |
| plot ID** | same ID for dry/wet run during one campaign | same ID for dry/wet run during one campaign | factorial | ID number |
| working group** | group ID | group ID | factorial | TUBAF/RISWC |

*The German KA5 classification system is used for soil texture (Sponagel and Ad-hoc-Arbeitsgruppe Boden (2005))

**Variables representing random effects in the used linear mixed effect models.





the sand and silt content and between the vegetation cover and time of consolidation. The sand content was removed as the soil texture predictor has less of a correlation with the skinfactor than the silt content. The time of consolidation was removed as a parameter because it is harder to obtain for model users than the vegetation cover. Among the factorial variables, the significant

impact that soil saturation (dry/wet experiments) has on the skinfactor was detected, which corresponds to the correlation between the skinfactor and the initial soil moisture. Dry soil leads to lower skinfactors than saturated soils. However, it is important to consider the soil saturation not only in the context of the soil moisture (low x high), but also in the context of the state of the topsoil. While dry experiments represent the natural conditions of the soil cover, wet experiments represent the soil cover after rainfall and impacts from the destruction of soil aggregates and soil crust, loss of trapped air, or water

repellence. The crop type and soil texture group also have an impact on the skinfactor, but only on the inter-level stage. For the crop predictor, unlikely relations were observed. Differences between similar crop groups (e.g., catch crops versus cereals) were more significant than the differences between highly diverse crop groups (e.g., catch crops versus seedbed). Significantly different skinfactor values were also observed between working groups.

To determine the transfer functions for the skinfactor, linear mixed-effect models (Galecky and Burzykowski, 2013) were

applied. All numerical soil input parameters and categorical variables used in previous studies were included in the analysis as fixed effects. Furthermore, two nested random effects were included in the model to account for the interdependency and hierarchy of the data. The first random effect is the working group. Results of the experiments can be affected by the use of a specific rainfall-runoff simulator. The rainfall parameters and methodology for data acquisition differ between the working groups (Table 2). The second random effect is the plot ID, which is nested in the working group. Both working groups usually

repeat their measurements twice on an identical plot to obtain data under the dry and wet conditions. Measurements with the same plot ID are thus interdependent.

## 2.4   Model selection

The experimental dataset was divided into the training subset, containing 75% of the randomly selected experiments, and validation subset, containing the remaining 25% of the experiments. Various models were fitted using the training subset.

Model ORIG, with factorial predictors originally used in the Parameter Catalogue, was fitted to statistically evaluate the current skinfactor prediction method available for model users (Michael et al., 1996). The dataset structures used in the Parameter Catalogue and presented in this study are not identical; therefore, the equivalents of the predictors were used to remain as close to the Parameter Catalogue approach as possible (e.g., factorial predictor plant development is not available for RISWC data; therefore, it was substituted by the numerical variable, vegetation cover). STEP1–STEP3 represent the group of models

manually selected using the stepwise method from the initial model containing all factorial predictors in the interactions with all numerical predictors. The manually controlled backward elimination approach was followed. Single predictors with the lowest significance were continuously removed from the model while controlling for the significance of the remaining predictors and interactions, the Akaike Information Criterion (AIC) (Akaike, 1987) and the environmental sensitivity of the selected predictors. STEP1 was the most complex model, whereas STEP2 and STEP3 were selected by simplifying model STEP1 to



provide a suitable model for EROSION-2D/3D users according to information on the study area and available predictors. The

simplest model, i.e., STRONG, contains only the two most significant predictors.

## 2.5 Prediction validation

Statistical reliability of the fitted models was measured based on the validation dataset, consisting of the remaining 25% of

the experimental data. In the first step of validation, skinfactors were predicted by transfer functions and compared to the

experimentally derived skinfactors. In the second step, an error propagation of the predicted skinfactors for surface runoff and

sediment volume was analysed. Soil and vegetation conditions from the validation datasets were applied on a hypothetical

400 m long and 9% steep slope. Surface runoff and sediment volume simulated with the experimentally derived skinfactor

was compared to the simulated skinfactor results. The goodness-of-fit of the measured and predicted skinfactor values was

evaluated with commonly used indicators: coefficient of determination ($R^2$), root mean square error (RMSE), mean absolute

percent error (MAPE), and the ratio of the RMSE and the standard deviation of the measured data $STDEV_{obs}$ (RSR). MAPE

works best if there are no extremes or zeros in the dataset. According to (Moriasi et al., 2007), model performance is satisfactory

if RSR <0.7, good if RSR <0.6, and very good if RSR <0.5. The last step of the validation was performed on real data collected

on three 40 cm * 50 cm plots equipped with rainfall gauges, runoff trap devices, and soil moisture meters. The plots were

placed in a field of oilseed rape, two in the middle of the slope, one in the upper part of the slope. During the 2017 vegetation

season, six rainfall events produced runoff. However, runoff was never recorded in all three plots, which shows high variability

in the rainfall-runoff processes even within a very small area. The parameters of the events are presented in Table 3. Each

rainfall event was modelled by Erosion-3D with the skinfactor predicted by transfer functions STEP1–3 and STRONG; for

each function, the skinfactor was corrected by the positive and negative MAPE error to account for the uncertainties in the

predictions.

**Table 3.** Rainfall events used for the skinfactor validation.

| date | initial moisture [%] | runoff volume [ml] | precipitation [mm] | max intensity [mm/5 min] | length [min] | saturation | comment |
|------|------|------|------|------|------|------|------|
| 05.05. | 28 | 0 - 20 | 4.4 | 0.6 | 50 | dry | |
| 14.05 | 27 | 0 - 100 | 12.8 | 7.4 | 390 | dry | |
| 29.06. | 24 | 0 - 160 | 19 | 1 | 320 | dry | crust |
| 02.07. | 38 | 0 - 40 | 3.2 | 0.4 | 190 | wet | crust + wet |
| 11.07. | 28 | 0 - 30 | 3.2 | 0.2 | 180 | dry | crust |
| 15.07. | 30 | 0 - 120 | 14 | 5.8 | 245 | wet | crust |

Saturation dry or wet was decided according to antecedent precipitation.





## 3 Results

### 3.1 Skinfactor prediction

Five models were fitted to evaluate the skinfactor estimation method given in the Parameter Catalogue and determine new
transfer functions for predicting skinfactors using the most significant predictors (Fig. 1). Table 4 lists the evaluation of
the model performance based on the validation dataset and the model predictors with the coefficients for transfer function
construction. The ORIG model, fitted to the predictor equivalents from the Parameter Catalogue, has low explanatory significance
(variance explained by fixed effects $R^2 = 0.12$). Only soil saturation (dry or wet experiment) is a highly significant predictor.
The new transfer functions provide significant improvement to the accuracy of skinfactor prediction. Soil moisture and bulk
density were determined as by far the most significant predictors, explaining together 79% of the skinfactor variability. The
skinfactor increased with an increase in both of the predictors (Fig. 2). Other significant predictors, e.g., silt content, soil texture
group, and soil saturation, slightly improved the model fit. The most complex STEP1 model containing all of the significant
predictors, including the interactions, explains only an additional 4% of the skinfactor variability. All four transfer functions
performed well according to the interpretation of the RSR indicator by (Moriasi et al., 2007). The mean absolute percent error
was between 66% and 72% for the new transfer functions while it was 192% for the ORIG function.

### 3.2 Error propagation

Error propagation of the predicted skinfactor for the surface runoff and sediment volume simulated by EROSION-3D was
evaluated on the hypothetical 400 m long slope. The skinfactors input into ORIG model produced no runoff for 24 out of the
64 validation datasets while the skinfactors input into the new transfer functions produced no runoff only for 1–3 datasets (Fig.
25 3). There is not a large difference in the error propagation between models STEP1–3 and STRONG (Fig. 5). This indicates the
major impact of the two strongest predictors, i.e., initial soil moisture and bulk density. The median error of the surface runoff
was 44–46% while that of the sediment volume was 52–56% (for the ORIG model these were 93 and 100%, respectively).
Errors below 100% characterised 78% of the datasets for surface runoff and 70% of the datasets for sediment volume, whereas,
for the ORIG model, these values were 50 and 42%, respectively. Table 5 statistically compares the model performance. STEP1
30 was the best performing model for both the surface runoff and sediment volume prediction (as compared with ORIG in Figs. 3
and 4). The simplest model, i.e., STRONG, produced better results for certain metrics than more complex models. In general,
all of the new transfer functions showed similar error propagation values, such that they can be used to predict the skinfactor.
The results suggest that the simplest function does not necessarily lead to the poorest result.

### 3.3 Validation with real events

Real rainfall-runoff events were modelled using the new transfer functions. To account for the potential error in the functions,
5 each event was simulated with the predicted skinfactor and the skinfactor corrected by +MAPE error and -MAPE error.
EROSION-3D simulated no runoff for four out of six the events using all of the transfer functions. Simulations with the





**Table 4.** Linear mixed effects models for skinfactor prediction: model evaluation based on the validation dataset using common statistical indicators, model variables, and their coefficients.

| | ORIG | STEP1 | STEP2 | STEP3 | STRONG |
|---|---|---|---|---|---|
| $R^2$ | 0.12 | 0.83 | 0.8 | 0.8 | 0.79 |
| RMSE | 2.11 | 0.91 | 1 | 1 | 1.02 |
| RSR | 0.94 | 0.41 | 0.45 | 0.45 | 0.46 |
| MAPE | 1.92 | 0.72 | 0.66 | 0.67 | 0.7 |
| Intercept | −3.6498 | −31.7377 | −17.3803 | −17.678 | −16.6319 |
| Initial soil moisture | — | 0.2845 | 0.1857 | 0.1711 | 0.1735 |
| bulk density | — | 0.0126 | 0.0072 | 0.0074 | 0.0074 |
| silt | — | 0.0847 | 0.0158 | 0.0195 | — |
| vegetation cover | $3 \times 10^{-4}$ | — | — | — | — |
| soil saturation- wet | 1.5319 | −1.864 | −0.3461 | — | — |
| soil texture class- sandy | −0.0761 | 20.7123 | — | — | — |
| soil texture class- silty | 0.4296 | 13.679 | — | — | — |
| type of management practice- conventional tillage | −0.179 | — | — | — | — |
| type of management practice- no tillage | −0.0256 | — | — | — | — |
| type of crop- cereals | 1.8503 | — | — | — | — |
| type of crop- erosion permitting crop | 1.3655 | — | — | — | — |
| type of crop- legume | 1.4049 | — | — | — | — |
| type of crop- oilseed crop | 0.3839 | — | — | — | — |
| type of crop- seedbad | 1.6929 | — | — | — | — |
| wet soil saturation : silt | — | 0.0238 | — | — | — |
| wet soil saturation : initial soil moisture | — | −0.0847 | — | — | — |
| wet soil saturation : bulk density | — | 0.0018 | — | — | — |
| sandy soil texture class : silt | — | −0.1055 | — | — | — |
| silty soil texture class : silt | — | −0.0679 | — | — | — |
| sandy soil texture class : bulk density | — | −0.0095 | — | — | — |
| silty soil texture class : bulk density | — | −0.0059 | — | — | — |
| sandy soil texture class : initial soil moisture | — | −0.0452 | — | — | — |
| silty soil texture class : initial soil moisture | — | −0.046 | — | — | — |

— indicates not included in the model.

A:B indicates interaction between factors A and B.

An example of transfer function construction (STEP3): $skinfactor = e^{-17.678 + 0.0195 * silt + 0.0074 * bulkdensity + 0.1711 * initialsoilmoisture}$





**Figure 1.** Experimentally derived versus predicted skinfactors (log values) for the validation dataset.





**Figure 2.** The dependency of the skinfactor on the bulk density and soil moisture. Point data represent whole dataset with experimentally derived skinfactors. Line data represent skinfactor prediction by STRONG for three different initial soil moisture conditions. ISM = initial soil moisture.





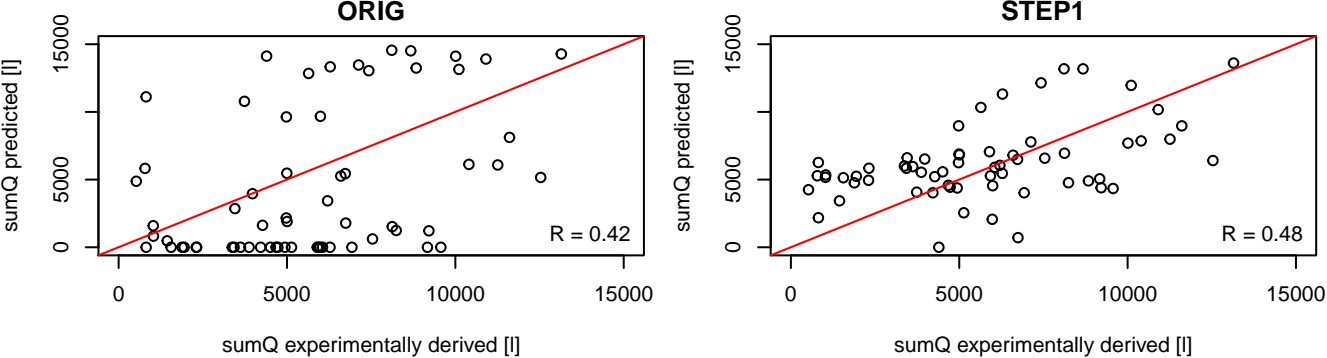

**Figure 3.** Surface runoff simulated with the derived skinfactor versus the ORIG skinfactor (left) and STEP1 skinfactor (right).

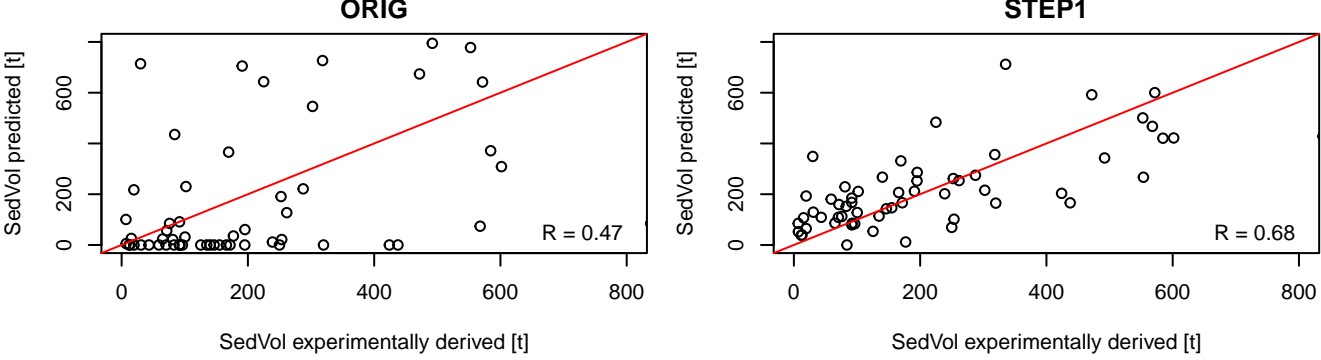

**Figure 4.** Sediment volume simulated with the derived skinfactor versus the ORIG skinfactor (left) and STEP1 skinfactor (right).

skinfactor corrected by -MAPE to increase the infiltration rate, produced no runoff for all events. Only events 14.5. and 15.7. produced runoff (Table 3). For all of the transfer functions, the modelled runoff was within the range or close to the runoff value recorded by the trap devices. The STRONG model simulated less runoff than the other models and only the simulations

10   with +MAPE correction produced runoff. The recorded runoff values for events 5.5., 2.7., and 11.7. are questionable, because the rainfall data had very low volume and intensity, significantly lower than the erosion causing rainfall, as defined by (Janeček et al., 2012) (12.5 mm volume or 6 mm/15 min intensity). Event 29.6. had one of the highest volumes, but had a relatively long duration and low intensity. While this event produced the largest runoff, as recorded by a trap device, EROSION-3D simulated no runoff. Crust on the topsoil was recorded by field workers for the last four events, which likely initiated runoff from the low-volume and low-intensity rainfall events. The fact that runoff was never recorded in three trap devices during

5   the same event shows the high natural variability of the rainfall-runoff process within a small area. More validation datasets for testing EROSION-3D under variable soil and vegetation conditions are required to properly validate the transfer functions.





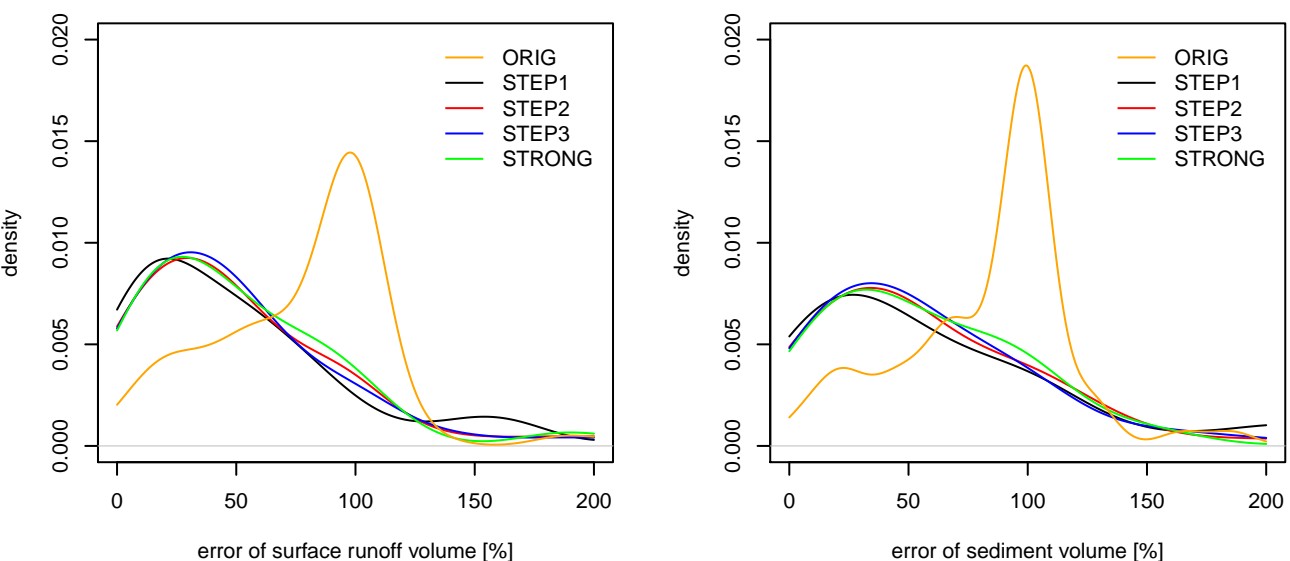

**Figure 5.** Error propagation for skinfactor prediction in the surface runoff (left) and sediment volume (right), a density plot of the percent error. Outlying experiments (error > 200%) create 6–9% of the validation experiments. Experiments with no simulated runoff is evaluated as 100% error.

Validation at the field or the catchment scale is appropriate because the measured runoff data represent average conditions, where site-to-site changes, as recorded using the trap device, are blurred.

### 3.4 Discussion

10 The joint rainfall simulation dataset of TUBAF and RISWC provides a sufficient amount of data to statistically analyse the relationships between the skinfactor calibration parameter and commonly measured soil and vegetation parameters, as well as to derive the transfer functions for the skinfactor.

  The current skinfactor prediction method published in the Parameter Catalogue is based on easily and accurately measurable factorial variables, i.e., crop, management practice, soil saturation, development stage of vegetation, and soil texture class. The results of this study show that the variables, except for the soil saturation have statistically negligible evidential influence on the skinfactor. The most significant predictors identified in this study, i.e., the initial soil moisture and bulk density, are highly variable in time and space and cannot be easily obtained. The initial soil moisture can be calculated from antecedent precipitation (Heggen, 2001) and other soil, vegetation, and relief properties (Pan et al., 2003; Zhao et al., 2011). Tramblay

5 et al. (2011) used external software to derive initial soil moisture as an input parameter for the runoff model. The number of projects developing methods and producing open data for soil moisture based on remote sensing techniques is increasing (e.g.,





**Table 5.** Error propagation of the skinfactor prediction models for the surface runoff and sediment volume evaluated by commonly used statistical indicators.

|  | ORIG | STEP1 | STEP2 | STEP3 | STRONG |
|---|---|---|---|---|---|
| surface runoff prediction | | | | | |
| no runoff simulated | 24 | 1 | 3 | 2 | 3 |
| outliers (error > 200%) | 4 | 5 | 6 | 6 | 5 |
| $R^2$ | 0.17 | 0.23 | 0.22 | 0.2 | 0.23 |
| RMSE | 5077 | 3077 | 3259 | 3311 | 3361 |
| RSR | 1.61 | 0.98 | 1.03 | 1.05 | 1.07 |
| MDAPE* | 0.93 | 0.45 | 0.44 | 0.46 | 0.45 |
| sediment volume prediction | | | | | |
| $R^2$ | 0.22 | 0.46 | 0.44 | 0.42 | 0.47 |
| RMSE | 288 | 174 | 184 | 188 | 187 |
| RSR | 1.24 | 0.75 | 0.79 | 0.81 | 0.8 |
| MDAPE* | 1 | 0.52 | 0.52 | 0.53 | 0.56 |

MDAPE: median absolute percent error. The median, instead of the mean, was used because of zero runoffs and outliers.

**Table 6.** Runoff volume [mL] from real rainfall events, measured versus simulated with the skinfactors predicted by the new transfer functions.

| date | measured sumQ | STEP1 sumQ | STEP2 sumQ | STEP3 sumQ | STRONG sumQ |
|---|---|---|---|---|---|
| 14.05 | 0 - 100 | 0 / 13 / 122 | 0 / 13 / 115 | 0 / 33 / 145 | 0 / 0 / 83 |
| 15.07. | 0 - 120 | 0 / 108 / 271 | 0 / 0 / 148 | 0 / 0 / 143 | 0 / 0 / 22 |

Measured sumQ: min - max value measured in three trap devices. Predicted sumQ: predicted - MAPE error / predicted / predicted + MAPE error.

soil moisture CCI data by ESA (Gruber et al., 2019) and soil moisture active passive (SMAP) data by NASA (Enrekhabi et al., 2014)). Copernicus ERA5-Land provides soil moisture data produced by the combination of model data with observations from across the world (Copernicus Climate Change Service (C3S), 2019). Bulk density can be estimated by pedotransfer functions based on the soil texture and organic carbon content. Sevastas et al. (2018) presented a review and validation of 56 pedotransfer functions found in the literature. Another review of direct and indirect estimation methods for bulk density was presented by Al-Shammary et al. (2018). Global maps of bulk density at various resolutions developed within the SMAP project are available in Das (2013). Ballabio et al. (2016) presented the European map of bulk density.

The relationship between the skinfactor and the soil moisture and bulk density indicates that infiltration rates are overestimated at low soil moisture and low bulk density values and underestimated at high bulk density values by the infiltration module used





in EROSION-2D/3D. Previous studies have also discussed the dependency on both the soil moisture and bulk density. Soil moisture has been explained by the stability of aggregates (Michael, 2000). Dry aggregates are prone to destruction by enclosed air, which becomes compressed by water infiltrating into the aggregates. The smaller particles from the destroyed aggregates then cause surface sealing and smaller skinfactors. Wet aggregates are more stable because their matrix potential is lower and the infiltrating water does not produce such high, destructive pressure in the aggregates. Schindewolf and Schmidt (2012) used air trapping on a larger scale. Air trapping occurs when the wetting front enters the soil. The enclosed soil air then hinders, to a certain extent, the infiltration. A further theoretical explanation is hydrophobicity, which results from hydrophobic particles (mainly organic matter) in the soil matrix. Once dried, particles are harder to rewet than hydrophilic particles (Hallett, 2007; Seidel, 2008; Kuhnert, 2008; Schindewolf, M.; Schmidt, 2009). All of these effects would decrease the infiltration rates for dry soils, but are not considered in model algorithms. Therefore, all of these theories can be considered reasonable explanations for the dependency of the skinfactor on soil moisture, but none of them are validated in the rainfall experiments. An alternative explanation is the misfit of the empirical estimation functions for the saturated hydraulic conductivity and matrix potential. The experimental basis behind Campbell's model is unknown (Campbell, 1985). The equations for the matrix potential estimation are based on the measurements of 40 important Belgian soil series. They represent a local dataset, which may be comparable to other regions, but validation is required (Vereecken, H. J., Maes, J., Feyen, J., and Darius, 1989). Schmidt (1996) showed that these equations lack accuracy for very dry conditions (pF >4).

The existing prediction methods for the skinfactor fail to include this dependency on soil moisture. They distinguish only between dry and wet run conditions (Michael et al., 1996; von Werner, 2009), which can rather correspond with the impact of the rainfall on soil cover (e.g., soil sealing and broken aggregates) as opposed to moisture (Fiener et al., 2011).

The dependency of the skinfactor on bulk density is associated with macropores and surface sealing (Michael, 2000). Soils with high bulk density values are likely treated with reduced tillage practice and therefore are rich on macropores, which enhance infiltration and lead to greater skinfactors. Soils with low bulk density are likely freshly ploughed and therefore are prone to surface sealing, which hinders infiltration and leads to lower skinfactors.

Previous studies associated skinfactor values greater than one with macropores and values smaller than one with surface sealing (Michael, 2000; Seidel, 2008; Schindewolf and Schmidt, 2012). This study indicates that skinfactor values do not systematically relate to these conditions; all of the experiments at dry conditions had skinfactors smaller than one, including those with reduced tillage, which tend to develop more macropores. However, this cannot be proven because surface sealing or macropore conditions were not recorded. Previous studies have attempted to determine the empirical equations for skinfactor prediction (Michael, 2000; Schlegel, 2012). Although these authors do not recommend the use of these equations (Schlegel, 2012), as well as the fact that certain predicted values are unreasonable (Lenz et al., 2018), the initial soil moisture and bulk density were identified as the most important predictors, which consistent with this study. In these attempts, experiments were grouped into subsets based on texture, management practices, and the type of run to derive regression models for the subsets. This method reduces the number of experiments and achieves higher $R^2$ values for each single subset, as compared with the method applied in this study, which uses categorical variables as covariables in linear models. Previous studies determined





different dependencies for the prediction parameters (e.g., the intercept of soil moisture) on each single subset, whereas this
study assumed an equal dependency on each parameter for the entire dataset.

## 4    Conclusion

This study aimed to increase the accuracy of the infiltration module of the EROSION-2D/3D soil erosion simulation tool by
introducing new transfer functions to estimate the skinfactor calibration parameter. The relationship of the skinfactor with soil,
vegetation, and farm management parameters was analysed using the linear mixed effect models based on 273 rainfall-runoff
experiments. The initial soil moisture and bulk density were found to be the most important predictors, together explaining 79%
of the skinfactor variability. These parameters are not considered in currently available prediction methods provided in (Michael
et al., 1996). Other significant predictors of soil texture (i.e., the silt content and KA5 soil texture group) and the impact of
previous rain events only slightly improved the skinfactor prediction. Four transfer functions with different complexities and
number of predictors were presented, such that the users can make a selection according to the available data in their study
area. The proposed transfer functions present significant increases in the skinfactor prediction accuracy, as compared with
currently available methods (decrease in the MAPE error from 192 to 66–72%). Error propagation of the estimated skinfactors
indicates substantial improvements to surface runoff and soil loss simulations. Real rainfall-runoff events were modelled by
EROSION-3D with the skinfactors predicted by the proposed functions, exhibiting good model performance for events with
higher total precipitation and intensity.

. This paper was compiled using the RMD-template by Nuest (Allaire et al., 2020) in RStudio (RStudio Team, 2020). The source file with
all calculations performed in R (R Core Team, 2020) and not open accessible input data are available in the supplementary materials.

. AR, JD, MM, and AB made rainfall experiments, HB, JL and JD processed rainfall experiments data, JL automatized skinfactor determination,
HB, JL and JD made the statistical analysis, IG provided data for validation on real events, HB and JL wrote the code and prepared manuscript,
AR and JK consulted the whole process.

. The authors declare no competing interests.

. This study was supported by the Ministry of Agriculture of the Czech Republic (QK1810341, MZE-RO0218) and by the European Social
Fund in the Free State of Saxony (Förderbaustein: Promotionen)



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
