# Peer review of "Improved calibration of Green-Ampt infiltration module in the EROSION-2D/3D model using a rainfall-runoff experiment database"

_SOIL, 2020_

## Referee Comment (RC1) · Mehdi Rahmati (Referee) · 26 Nov 2020

Review Report on Manuscript "soil-2020-62" entitled "Improved calibration of Green-Ampt infiltration in the EROSION-2D/3D model using a rainfall-runoff experiment". The paper is dealing with infiltration submodule of the EROSION-2D/3D model. Authors are trying to provide some pedo-transfer functions (PTFs) to optimize the prediction of the needed calibration factor in infiltration submodule, named skinfactor. The skinfactor can be determined experimentally from rainfall-runoff or infiltration experiments with the hillslope simulation tool EROSION-2D. However, authors believe that the process is time consuming, labor-extensive, and limited from several aspects. So, the determination of the skinfactor may restrict the usage of the EROSION-2D/3D model in the cases where the experimentally data of rainfall-runoff or infiltration are missed. So, the parameter catalogue of the model provides extra studies to predict skinfactor from several readily-available parameters. However, authors believe that the studies are limited and provide data for limited conditions. Authors also criticize the provided regression models for their weak determination coefficients. Therefore, authors provide a R package enabling automated and batch determination of the skinfactor for, what they called, an extensive rainfall-runoff infiltration experiment dataset. They used database containing 464 experiments only, from which around 191 experiments are excluded from further analysis. Overall, I found the study very interesting and worthwhile. However, some questions arise when reading the paper: 1) The first question is that what is the reason authors have limited themselves to limited number of the experiments while they themselves are criticizing the model's provider providing skinfactor prediction with limited number of experiments (I think, 116 experiments). Since the skinfactor is predictable from both rainfall-runoff and infiltration experiments, why not to use infiltration experiments which are available in higher numbers. For example, SWIG database (Rahmati et all. 2018 , https://essd.copernicus.org/articles/10/1237/2018/) provides more than 5000 infiltration experiments (including 374 rainfall simulator experiments) from all around the world that can be used to provide a global PTF for skinfactor predictions. 2) The second question is that why the linear fixed-effect model is used to develop the PTF and why nonlinear methods (or let say machine learning methods) are not used? I understand that linear mixed-effect method is much stronger than simple regression methods since it accounts for both explained and non-explained variations in independent variable. However, the relationship between skinfactor and soil readily-available parameters seems more nonlinear to me and I think machine learning method may act much better than the linear mixed-effect method. 3) I believe that the reason why the working group plays an important role (as input parameter) in prediction of skinfactor is that the used database is not global enough. I believe if we use a larger database, we can simply provide a global PTF being free from working groups effects. 4) In the

MM section, please clarify that how you have determined experimentally the skinfactor for PTFs development. 5) In model selection section, I see authors have correctly divided the database into two groups of training and validation subsets. However, there should be one more step to assess the reliability of the developed PTFs. What authors have done is only assessing the accuracy of the models. However, the accuracy may be rooted in chance since you divide the database into training and validation subsets randomly. So, I suggest authors to repeat the process of splitting data into training and validation subsets 10 times (at least) and calculate the criterions. Finally, check the STD between obtained results of 10 times. It will give you a better understanding of the reliability of the PTFs.

Sincerely Yours, Mehdi Rahmati

---

## Author Comment (AC1) · 7 Dec 2020

Dear Mehdi Rahmati, thank you very much for your insightful comments on our manuscript "Improved calibration of Green-Ampt infiltration in the EROSION-2D/3D model using a rainfall-runoff experiment database". Besides your suggestions 4 and 5, which we find very worthwile and will consider them in adjusted manusript after the discussion will be completed, we would like to respond to your comments 1-3.

1) You ask, why did we use only limited database of rainfall-runoff experiments and why we did not use infiltration experiments, which are avilable in higher amount. The skinfactor can be basicaly derived only from rainfall-runoff experiments as the rainfall

duration and intensity are input parameters in the skinfactor determination. We see, that the name rainfall-runoff and infiltration experiments is missleading and we suggest to correct it in the manusript. Except this, using pure infiltration data would likely raise further issues, as experimental methodology is completely different and also some differing processes are simulated with the infiltration experiments in contrast to rainfall-runoff experiments (permanent submergence; no simulation of aggregate destruction and surface sealing, ...). Reason for our limited database is, that the skinfactor determination requires lots of input parameters which are rarely all measured during rainfall-runoff experiments. Our first intention was to use a database published by Seibert et al. (2011, DOI: 10.1594/GFZ.TR32.2) containing 726 simulations from European countries, however, we had to exclude all experiments except those made by model developpers due to missing parameters. Similarly, non of the experiments in the suggested SWIG database includes full set of input parameters. We suggest to highlight the spatial limination of the PTFs in the manuscript and comment, why those databases could not be used in our work.

2) You ask, why the linear mixed-effects model is used to develop the PTFs and why nonlinear methods (machine learning methods) are not used. We agree, that relationships between the parameters are probably more complex than linear mixed-effects models can cover. Using machine learning methods, however, would not give us an insight in the relationships between skinfactor and variables, which is an important part in our analysis. Linear-mixed effects models allow us to describe the relationships clearly by coefficients so the PTFs are easy to interpret and can be discussed in the context of the rainfall-runoff processes.

3) You believe, that the reason why the working group plays an important role in prediction of skinfactor is that the used database is not global enough. In general we agree that your assuption can be correct. The reasons why our database is not global is discussed in comment one. In case of our database it was important to consider the impact of working group.

---

## Referee Comment (RC2) · Anonymous Referee #2 · 7 Feb 2021

This paper uses a large set of rainfall-runoff experiments to derive relations between a parameter of the Green-Ampt infiltration model and other soil properties, states and soil management parameters. A linear mixed effects model is used to setup these relations. The estimated parameters with this mixed model are subsequently validated against a validation set of run-off experiments. In a second step, the propagation of errors in predictions of run-off and sediment load are evaluated.

The paper addresses a relevant topic and makes an important contribution to bringing together datasets that can be used to develop pedotransfer functions for the saturated hydraulic conductivity for which reliable pedotransfer functions are still missing. The

reason is that this parameter is spatially but also temporally very variable and depending on soil structural properties that cannot be quantified easily.

The authors did not determine the saturated hydraulic conductivity but a scaling factor, the skinfactor, that scales the saturated hydraulic conductivity, is estimated. This is an interesting approach since it allows to evaluate the impact of certain soil properties on the correction of an a-priori estimate of the saturated hydraulic conductivity. But, the authors should pay more attention to the background and knowledge of the readers. The skinfactor is a specific parameter of a certain model. Neither this model nor the meaning of this parameter are known to most of the readers. As it scales the saturated conductivity, it is also crucial to explain how the saturated hydraulic conductivity was derived. The authors refer to the 'Campbell estimation' but do not give a reference and do not explain how this model estimates the saturated hydraulic conductivity and based on which parameters.

The authors should best give the equations that are solved in the infiltration model and explain the different parameters that are used. It is for instance not clear to me how they determined the matrix potential at the wetting front, which is an important parameter in the Green-Ampt equation. Since they write about estimating the soil water potential of the dry soil using pedotransfer functions, I suspect that they used the dry soil water potential as the water potential at the wetting front. This is in fact an incorrect interpretation of the Green-Ampt model parameters since the water potential at the wetting front is only a very week function of the water potential of the dry soil. See Dingman S.L, Physical Hydrology, 1993. Using the water potential of the dry soil as a proxy for the water potential at the wetting front would lead to an overestimation of the infiltration rate and this overestimation could explain why the authors found skin factors < 1 for dry soils. If the authors did not estimate the water potential at the wetting front from the water potential in the dry soil, they should mention how they estimated this parameter and based on which soil properties. Finally, the saturated water content is also a parameter of the Green-Ampt model. The authors do not provide information

about how this parameter was estimated.

In general, I think that the authors should describe more clearly the methods they used and give more information to the readers. At several places, the authors are implicitly referring to background information that is not available to the readers (examples are given below). Also in the results and discussion section, the authors are referring to results that have not been presented yet. So I think that the structure of the paper requires some improvement as well as the language, which is at some places confusing or contradictory.

Detailed comments Title: You cannot 'calibrate' infiltration. You calibrate a model or parameters of a model. I think you should make clear that you estimated a scaling factor of the saturated hydraulic conductivity, which was estimated with the Campbell equations.

P3 ln 9: I do not understand the role of the matrix potential in this context and how the matrix potential can be estimated since it is not a static soil property. It is tempting to interpret the soil matrix potential in the dry soil as the matric potential at the wetting front. But this is an incorrect interpretation of the matrix potential at the wetting front. The matric potential at the wetting front is in fact independent of the antecedent soil moisture.

P3 ln 16: These are very strange units of the saturated conductivity. Normally saturated conductivity is expressed in m s-1.

P3 ln 24: I am wondering how the skinfactor is derived from the infiltration rate at the end of the experiment. If the infiltration experiment lasts long enough, then the infiltration rate converges to the saturated hydraulic conductivity and the skinfactor can be derived directly from the infiltration. In fact, no Green Ampt infiltration model is needed then to derive the parameter. The authors should be more explicit on how they derived the skin factor from the infiltration rate at the end of the experiment. Did they use the Green Ampt infiltration model or not? How did they decide that the infiltration

rate did not change over time anymore? Another question is how were the initial and saturated soil moisture content defined and what was the pressure at the wetting front?

P5, table 2: You used time of consolidation as a predictor variable. But, also relevant for consolidation is the cumulative precipitation after the last topsoil disturbance.

P6 ln 6: 'Dry soil leads to lower skinfactors than saturated soils' This comes a bit unexpected since no results have been shown yet.

P6 ln 8: 'While dry experiments represent the natural conditions of the soil cover, wet experiments represent the soil cover after rainfall and impacts from the destruction of soil aggregates and soil crust, loss of trapped air, or water repellence.' How is this related to the difference between the skin factors for dry and wet experiments?

P6 ln 10: 'The crop type and soil texture group also have an impact on the skinfactor, but only on the inter-level stage.' What is inter-level stage?

P6 ln 25: You must explain which variables are used in the 'Parameter Catalogue'. This catalogue is probably not known to many readers.

P6 ln 33: Explain what you mean with 'environmental sensitivity'.

P6 ln 34: You must give more information about how STEP1 was simplified and what the difference is between STEP2 and STEP3.

P6 ln 14: Is there actually a difference between RSR and the square root of the coefficient of determination?

P7 ln 16: Give information about the location of the site.

P12 Figure 3 and 4: It is not clear to me whether the runoff volumes and sediment 'volumes' were measured or were predicted using the experimentally derived skin factors. I think the latter is the case. What are 'sediment volumes' and why are they expressed in tons? That is not a volume unit.

P12: 'skinfactor corrected by -MAPE to increase the infiltration rate, produced no runoff' I did not understand this. If you add a negative number, then the skinfactor decreases and shouldn't the infiltration rate then decrease and more runoff be produced?

P13 ln 10: How did you decide that the dataset provided sufficient data?

P15: 'An alternative explanation is the misfit of the empirical estimation functions for the saturated hydraulic conductivity and matrix potential. The experimental basis behind Campbell's model is unknown (Campbell, 1985). The equations for the matrix potential estimation are based on the measurements of 40 important Belgian soil series.' I suppose that the matrix potential was calculated from the initial soil moisture content using the water retention curve and that the parameters of the water retention curve were derived from other soil information using pedotransfer functions. But it is not clear to me how this matrix potential was afterwards used in the Green-Ampt infiltration model. It is incorrect to assume that this is the water potential at the wetting front, hf. hf is related to the sorptivity of the soil, Ks, and the difference between the saturated and initial water content. The sorptivity of the soil depends on the water potential of the dry soil but only very weak. The sorptivity is the integral of the weighted unsaturated conductivity between the water potential of the dry soil and the pressure head at the soil surface (which is 0 in case of saturation). Since the unsaturated hydraulic conductivity decreases so strongly with more negative matrix heads, this integral is not very sensitive to the lower boundary of this integral (the matrix potential in the dry soil). As a consequence, the sorptivity and hf are not very sensitive to the matrix potential of the dry soil. Furthermore, hf varies from a few cm to about -30 cm in clayey soils. This is of course much less negative then the matrix potentials in dry soils. As a consequence, using the matrix potential of the dry soil as hf will lead to a strong underestimation of the pressure head at the wetting front and a strong overestimation of the infiltration that is driven by capillarity. In order to compensate for this overestimation of infiltration, the saturated conductivity must be reduced to match the measured infiltration rates. This may be the reason why the skin factors are reduced when the initial soil moisture

content is lower.

P15 ln 37: 'This method reduces the number of experiments' Do you mean: the number of parameters?

P 15 ln 38: 'Previous studies determined different dependencies for the prediction parameters (e.g., the intercept of soil moisture) on each single subset, whereas this study assumed an equal dependency on each parameter for the entire dataset.' But in the STEP1 model, you considered interactions between the categorical and continuous predictors.

P 16 ln 17: 'Other significant predictors of soil texture' You do not predict soil texture but you use soil texture as a predictor. 'Other significant predictors such as soil texture'

---

## Editor Comment (EC1) · Jan Vanderborght (Editor) · 13 Mar 2021

Dear Authors,

thank you for your detailed responses to the reviewers' comments. I think that the changes you are proposing will address most of the comments convincingly. However, I have concerns about the treatment of the water potential at the wetting front. As you describe in your replies, the initial water content is used to estimate the water potential at the wetting front in the EROSION2/3D model. But according to reviewer 2, this is not correct and leads to an adjustment of a lower skin factor for drier soils.

A lower skin factor would also imply a lower steady state infiltration rate. The figure that you are showing in your replies indeed suggests that you are understimating the steady infiltration rate. The simulated infiltration rates still decrease further whereas the measurements seem to have reached a constant value. I understand that this is the way it is implemented in EROSION2/3D but this would mean that the Green and Ampt model is incorrectly implemented in the EROSION2/3D. A consequence of this is that skin factors need to be adjusted as a function of the initial soil water content. But, this is only required when the Green-Ampt model is incorrectly implemented and cannot be transfered to other models that use a correct implementation of the Green-Ampt model. Furthermore, it would lead to an underestimation of the steady infiltration rate. In order to address the issue of the effect of the initial water content on the infiltration characteristic, I think it is essential that you show simulations with a 1D model that solves Richards equation, such as the freely available Hydrus1D code. With this code, you can simulate infiltration for different initial water contents using always the same hydraulic parameters. The infiltration curves that are simulated can then be fitted with a Green-Ampt model using a constant and known Ksat (in the Hydrus simulations, Ksat was not changed between simulations with different initial water content and so it shouldn't be changed in the Green and Ampt model either) by fitting the water pressure at the wetting front. Alternatively, you could fix the water potential at the wetting front in the Green and Ampt model to the water potential of the dry soil and fit the Ksat used in the Green Ampt model. I suspect that in that case, you will fit a smaller Ksat for drier soils (or a smaller skin factor) but the Green Ampt model will not be able to simulate the steady state infiltration rate, which is the same in all simulations and does not depend on the initial water content.

---

## Author Comment (AC3) · 17 Mar 2021

Dear Prof. Vanderborgh, thank you very much for your comment on our paper and the discussion upon it. We understand your concerns about the treatment of the water potential at the wetting front, however, we are sure, that the implementation of the Green-Ampt approach (GA) in EROSION-3D is correct as we show below. For numerical efficiency there are many different explicit implementations of GA, all bringing larger or smaller errors. Our intention was not to dig into the theory of GA (we agree that it would be a problem to choose a model where GA is wrongly implemented), but to improve the calibration of GA in the practical relevant context of a soil erosion

model. Our research shows an approach of predicting the calibration parameters from easily measurable parameters based on experimental data to systematically improve the model results. The possibility to analyze the error source is a benefit of the used statistical methods and one of the aims of the study. The fact that it can lead to detection of specific shortcoming in a model we see as an added value of the study and this discussion, not as reason to choose another model. On one hand it can trigger future development of the model and close the gap, on the other hand it makes the user aware of the model limits and gives him a way to compensate for it. EROSION-3D belongs among used and respected models and we find it reasonable to choose it for such study. At the same time we believe that the approach can be transferred to other models and lead to their development and achievement of better results.

We will fully respect your decision regarding publishing in SOIL. We understand if a more theoretical research would fit better to the journal scope. However we believe it has its place among scientific papers and will find interested readers.

Please, see more detailed explanation regarding the GA implementation in EROSION-3D.

As we tried to figure out in our response to reviewer 2, the water potential at the wetting front is not necessarily independent of the initial soil moisture. There might be cases in which this simplification is sufficient, but in general we don't think that the wetting front suction can only be a function of soil texture. As a matter of fact when completely following this suggestion, a given soil would be assigned with one value for wetting front suction and one value for saturated hydraulic conductivity, without considering any dependency on initial soil water content. The whole variance in modelled infiltration rates would then result from the difference of fillable pore space and initially water filled pore space. We don't think that this variance can sufficiently reflect the variance found in natural or simulated rainfall events. The still decreasing modelled infiltration rates, which can be seen at points where experiment already shows more or less steady conditions, are caused by the applied method of model calibration. By calibrating only hydraulic conductivity (and not matrix potential) a better fit from modelled to experimental curves is not achievable. From a comparison of the implicit example for the Green-Ampt model given by Dingman with the EROSION-3D algorithms we are pretty sure the Green-Ampt algorithms implemented in EROSION-3D are correct (see also attached figure). The explicit function used in EROSION-3D might not be the best approximation, but reflects the general characteristics of the Green-Ampt model sufficiently. Assuming that we use always the saturated hydraulic conductivity to run Green-Ampt models we would always limit the steady infiltration to this value. By doing so we would not be able to account for a reduced hydraulic conductivity under unsaturated conditions. If one input unsaturated hydraulic conductivity (with a lower value) this will be the value the modelled infiltration rate will asymptotically approach. For our understanding this is a general shortcoming of all Green-Ampt implementations, unless a dynamic adaptation of wetting front suction, hydraulic conductivity from initial unsaturated conditions toward saturated conditions is implemented. Regarding the Hydrus-1D model, we see the value of such a comparison but as explained above, this would lead away from our approach to use experimental data as a fitting target to a more theoretical asset of the Green-Ampt model. As we see the Green-Ampt implementation in EROSION-3D as not perfect, but valid, we prefer to stay on our intended research layout.

Regarding the two fitting strategies you propose in your last paragraph - Basically we followed the second one in our paper, to fix the water potential at the wetting front to the water potential of dry soils and fit the Ksat values via the skinfactor on our experimentally derived infiltration curves. Your suspicion that GA is then not able to model steady state conditions, is proved by the figure provided in response to reviewer 2.

On behalf of all co-authors, Yours sincerely Hana Beitlerová

[Figure]

**Fig. 1.** Soil infiltration curve simulated based on Dingman (2015) and EROSION-3D algorithms

---

## Author Response (AR1)

Dear editor and reviewers,

we see the parametrization issue opened by the second reviewer, value the quality of this review and want to thank you for your contribution in this discussion. This is opening a bigger topic in EROSION-2D/3D development, which can be accessed either through the algorithms implemented in the source code or through different methods of model parameterization. Both ways are beyond our current possibilities and have to be a topic of separate research.

We prepared a new version of the manuscript where the core of our study remains the same, but we discuss the parametrization issue. We tested the parameter optimization approach on the infiltration module for one rainfall experiment and compared it with the state-of-the-art method, as you already showed with the Hydrus infiltration curve. We finally propose to integrate this parameter fitting strategy in the future development of the model.

Other proposed changes are also reflected as stated in the point by point answers. Several sections (reproducibility of the analysis by R, initial selection of experiments, STEP3 model) of the paper are now seen less important and were therefore excluded in the revised manuscript.

We ask you for a quick consideration of the revised manuscript regarding acceptance as the funding for the authors (e.g. PhD scholarship) ended or will end soon and our projects do rely on this manuscript.

The point by point answers are copied from the author's comments posted during the open discussion. The corresponding changes in the manuscript are described.

Black – reviewer's comment
Green – author's respond
Red – implementation in the manuscript

**Point by point answers to review 1**

1) The first question is that what is the reason
authors have limited themselves to limited number of the experiments while they themselves
are criticizing the model's provider providing skinfactor prediction with limited
number of experiments (I think, 116 experiments). Since the skinfactor is predictable
from both rainfall-runoff and infiltration experiments, why not to use infiltration experiments
which are available in higher numbers. For example, SWIG database (Rahmati
et all. 2018 , https://essd.copernicus.org/articles/10/1237/2018/) provides more
than 5000 infiltration experiments (including 374 rainfall simulator experiments) from
all around the world that can be used to provide a global PTF for skinfactor predictions.

The skinfactor can be basically derived only from rainfall-runoff experiments as the rainfall duration and intensity are input parameters in the skinfactor determination. We see,
that the name rainfall-runoff and infiltration experiments is misleading and we suggest
to correct it in the manuscipt. Except this, using pure infiltration data would likely raise
further issues, as experimental methodology is completely different and also some
differing processes are simulated with the infiltration experiments in contrast to rainfall runoff
experiments (permanent submergence; no simulation of aggregate destruction
and surface sealing, ...). Reason for our limited database is, that the skinfactor determination
requires lots of input parameters which are rarely all measured during rainfall runoff
experiments. Our first intention was to use a database published by Seibert
et al. (2011, DOI: 10.1594/GFZ.TR32.2) containing 726 simulations from European
countries, however, we had to exclude all experiments except those made by model

developers due to missing parameters. Similarly, non of the experiments in the suggested SWIG database includes full set of input parameters. We suggest to highlight the spatial limiation of the PTFs in the manuscript and comment, why those databases could not be used in our work.

In Data section localisation of the data is mentioned, in discussion section the spatial limitation of the database is highlited, other existing databases are mentioned and explained why they could not be used.

2) The second question is that why the linear fixed-effect model is used to develop the PTF and why nonlinear methods (or let say machine learning methods) are not used? I understand that linear mixed-effect method is much stronger than simple regression methods since it accounts for both explained and non-explained variations in independent variable. However, the relationship between skinfactor and soil readily-available parameters seems more nonlinear to me and I think machine learning method may act much better than the linear mixed-effect method.

We agree, that relationships between the parameters are probably more complex than linear mixed-effects models can cover. Using machine learning methods, however, would not give us an insight in the relationships between skinfactor and variables, which is an important part in our analysis. Linear-mixed effects models allow us to describe the relationships clearly by coefficients so the PTFs are easy to interpret and can be discussed in the context of the rainfall-runoff processes.

3) I believe that the reason why the working group plays an important role (as input parameter) in prediction of skinfactor is that the used database is not global enough. I believe if we use a larger database, we can simply provide a global PTF being free from working groups effects.

In general we agree that your assumption can be correct. The reasons why our database is not global is discussed in comment one. In case of our database it was important to consider the impact of working group.

4) In the MM section, please clarify that how you have determined experimentally the skinfactor for PTFs development.

Added to the Methods section

5) In model selection section, I see authors have correctly divided the database into two groups of training and validation subsets. However, there should be one more step to assess the reliability of the developed PTFs. What authors have done is only assessing the accuracy of the models. However, the accuracy may be rooted in chance since you divide the database into training and validation subsets randomly. So, I suggest authors to repeat the process of splitting data into training and validation subsets 10 times (at least) and calculate the criterions. Finally, check the STD between obtained results of 10 times. It will give you a better understanding of the reliability of the PTFs.

Workflow changed according to the reviewer's suggestion

**Point by point answers to review 2**

R: Title: You cannot 'calibrate' infiltration. You calibrate a model or parameters of a model. I think you should make clear that you estimated a scaling factor of the saturated hydraulic conductivity, which was estimated with the Campbell equations.

A: Here we politely dare to disagree with your opinion. Green-Ampt is one of the broadly used infiltration models. We assume it is understandable from the title, that we calibrate a model, not the infiltration itself, however, we suggest to add the world module in the title to make it clear. We find that the title "Improved calibration of Green-Ampt infiltration module in the EROSION-2D/3D model using a rainfall-runoff experiment database" sufficiently expresses the content of this paper and more detailed information can be specified in the background section.

Improved calibration of Green-Ampt infiltration module in the EROSION-2D/3D model using a rainfall-runoff experiment database

R: P3 ln 9: I do not understand the role of the matrix potential in this context and how the matrix potential can be estimated since it is not a static soil property. It is tempting to interpret the soil matrix potential in the dry soil as the matric potential at the wetting front. But this is an incorrect interpretation of the matrix potential at the wetting front. The matric potential at the wetting front is in fact independent of the antecedent soil moisture.

A: At this point we must again disagree. To quote Dingman Physical Hydrology (3rd edition, 2015, p. 371): "In general, the wetting-front suction _f is a function of time, ponding depth, initial water content, and soil type." There might be situations in which an estimate of the matrix potential independent of initial (or antecedent) soil moisture is sufficient, but the independence is not a fact. So indeed EROSION-3D uses the initial soil moisture as input parameter in EROSION-2D/3D to estimate matric potential at the wetting front with the Van-Genuchten/Vereecken equations. We will provide the equations in the manuscript.

Equations added in methodology section, implementation of Green-Ampt algorithms and method of parameter optimization discussed in discussion section. Problematic mentioned in abstract and conclusion.

R: P3 ln 16: These are very strange units of the saturated conductivity. Normally saturated conductivity is expressed in m s-1.

A: The units are used by Campbell (1985), which is source of the calculations of Ks in EROSION-2D/3D model. According to Campbell (1985) an hydraulic conductivity of 1 kg * s /m3 equals 9.8 * 10^(-3) m/s (divided by water density and multiplied by the gravitational constant).

R: P3 ln 24: I am wondering how the skinfactor is derived from the infiltration rate at the end of the experiment. If the infiltration experiment lasts long enough, then the infiltration rate converges to the saturated hydraulic conductivity and the skinfactor can be derived directly from the infiltration. In fact, no Green Ampt infiltration model is needed then to derive the parameter. The authors should be more explicit on how they derived the skin factor from the infiltration rate at the end of the experiment. Did they

use the Green Ampt infiltration model or not? How did they decide that the infiltration rate did not change over time anymore? Another question is how were the initial and saturated soil moisture content defined and what was the pressure at the wetting front?

A: The infiltration curve of each experiment was simulated by EROSION-3D. Skinfactor was iteratively adapted, until the simulated and measured end infiltration of the experiment matched. The two established methods of skinfactor derivation are shown in Fig. 1 - upper dashed line is simulated infiltration to match the end infiltration of the experiment, lower dashed line to match the cumulative runoff. We only assume that the infiltration rate did not change over time any more. The experiments of TUBAF last until a steady runoff rate is reached, which is decided by the knowledge of the experts in field. The dry experiments of RISWC last 30 minutes, in most cases measured curves look like the blue one in the picture, where steady state seams to be reached. Wet experiments last 15 minutes, steady state seems to be reached after first few minutes. Initial soil moisture was measured in field for each experiment. Saturated soil moisture is calculated by Vereecken, 1989 and pressure head at wetting front (matric potential) by Van Genuchten, 1980. We will provide the equations and references in the corrected manuscript.

Equations added in methodology section, parametrization of skinfactor described and illustrated in figure, critically discussed in discussion section.

R: P5, table 2: You used time of consolidation as a predictor variable. But, also relevant for consolidation is the cumulative precipitation after the last topsoil disturbance.

A: We are aware about this, however, information on cumulative rainfall is not available for the experiments. An experimental site is usually prepared in spring and experiments are performed during whole vegetation season. No rainfall gauge to collect rainfall is installed in case of used experiments. As given on P6 ln 2 time of consolidation was removed from the statistics, because of autocorrelation with vegetation cover, which is easier to obtain for the model users and is an input parameter of EROSION-3D model, so the model user must have the information about it anyway.

In the new perspective on the revised manuscript relevant section was abridged. Time of consolidation was not used as predictor and is not mentioned in the text.

R: P6 ln 6: 'Dry soil leads to lower skinfactors than saturated soils' This comes a bit unexpected since no results have been shown yet.

A: We will work on the paper structure and move this to results or discussion.

The section was abridged

R: P6 ln 8: 'While dry experiments represent the natural conditions of the soil cover, wet experiments represent the soil cover after rainfall and impacts from the destruction of soil aggregates and soil crust, loss of trapped air, or water repellence.' How is this related to the difference between the skin factors for dry and wet experiments?

A: Initial soil moisture and the version of experiment - dry and wet can be seen as the same variable by the readers, just once given as numerical and once as two-level categorical variable. Therefore we found important to explain, that the dry/wet express rather difference in state of the soil, than the difference in the wettnes itself. If the

mentioned features - soil crust, trapped air, water repellency, are developed in the soil, the infiltration can be decreased significantly. However, none of the feature is considered in the infiltration module. It can play a role in the result of our analysis, that infiltration in dry soils is overestimated by the model and why skinfactors of dry soils are smaller than for wet runs.

In the new perspective on the revised manuscript relevant section was abridged. This topic is mentioned in discussion.

R: P6 ln 10: 'The crop type and soil texture group also have an impact on the skinfactor, but only on the inter-level stage.' What is inter-level stage?

A: It means, that overall the predictor has no significant impact on skinfactor, but significant difference can be found between some of the variable levels (categories). We will reformulate the part to be more clear.

Formulation changed

R: P6 ln 25: You must explain which variables are used in the 'Parameter Catalogue'. This catalogue is probably not known to many readers.

A: Table 4 contains the list of variables, we give a reference to the table here or will add the missing information directly in the text.

Mentioned in the text

R: P6 ln 33: Explain what you mean with 'environmental sensitivity'.

A: This is what the "manually controlled" backward elimination means. There are automatic model selection methods, however, the algorithms behind these methods do not know the studied processes. The variable selection can be correct mathematically and statistically, but it can be nonsense from the environmental point of view. There- fore we were deciding manually which variables are removed from the model, not only based on the significance and AIC, but also based on our knowledge of the infiltration process. The expression "environmental sensitivity" is confusing and we reformulate it.

Removed from text

R: P6 ln 34: You must give more information about how STEP1 was simplified and what the difference is between STEP2 and STEP3.

A: In this section we describe methodology. Our intention is to give the best model STEP1 and then more simple models to be easier to use for the model users, but at this point we do not know how they look like. The final selected models are given in table 4 in results. We will formulate the methods section more clearly and give more space to the model description in the results.

We formulated the methods section more clearly and give more space to the model description in the results.

R: P6 ln 14: Is there actually a difference between RSR and the square root of the coefficient of determination?

A: R2 describes the proportion of the variance in measured data explained by the model. RSR is calculated as the ratio of the RMSE and standard deviation of measured data. Root square of R2 is not equal to RSR.

R: P7 ln 16: Give information about the location of the site.

A: We will add the location in the revised paper.

Coordinated added

R: P12 Figure 3 and 4: It is not clear to me whether the runoff volumes and sediment 'volumes' were measured or were predicted using the experimentally derived skin factors. I think the latter is the case. What are 'sediment volumes' and why are they expressed in tons? That is not a volume unit.

A: They were simulated with the experimentally derived skinfactors as written in the title of the figures. You are right regarding the units. We correct it to sediment mass.

Corrected units

R: P12: 'skinfactor corrected by -MAPE to increase the infiltration rate, produced no runoff'. I did not understand this. If you add a negative number, then the skinfactor de- creases and shouldn't the infiltration rate then decrease and more runoff be produced?

A: MAPE was calculated for ln(skinfactor), which has negative numbers for skinfactor< 1. Therefore -MAPE correction leads to less negative numbers of ln(skinfactor), which result in higher skinfactor and thus lower surface runoff.

R: P13 ln 10: How did you decide that the dataset provided sufficient data?

A: We did not test any hard criterion to decide, but followed the basic statistical principles. There is many times higher amount of experiments than independent variables, skinfactor (log) has near normal distribution, same as most of numerical variables follow near normal or other expected distribution, categorical data were recategorized to avoid under-represented categories. Rainfall experiments data in general are of limited amount and presented calibration process requires input parameters, which are usually not all measured. Bigger open datasets exist, but all we examined did not have complete set of measured parameters for our purposes. This dataset contains more than twice more experiments, than the original dataset used in previous studies.

R: P15: 'An alternative explanation is the misfit of the empirical estimation functions for the saturated hydraulic conductivity and matrix potential. The experimental basis behind Campbell's model is unknown (Campbell, 1985). The equations for the matrix potential estimation are based on the measurements of 40 important Belgian soil series.' I suppose that the matrix potential was calculated from the initial soil moisture content using the water retention curve and that the parameters of the water retention curve were derived from other soil information using pedotransfer functions. But it is not clear to me how this matrix potential was afterwards used in the Green-Ampt infiltration model.

A: Indeed the algorithms of EROSION-3D follow the tempting approach to use soil

matric potential. For a better understanding of the infiltration algorithms implemented in the model the equations will be added to the revised paper.

R: It is incorrect to assume that this is the water potential at the wetting front, hf. hf is related to the sorptivity of the soil, Ks, and the difference between the saturated and initial water content. The sorptivity of the soil depends on the water potential of the dry soil but only very weak. The sorptivity is the integral of the weighted unsaturated conductivity between the water potential of the dry soil and the pressure head at the soil surface (which is 0 in case of saturation). Since the unsaturated hydraulic conductivity decreases so strongly with more negative matrix heads, this integral is not very sensitive to the lower boundary of this integral (the matrix potential in the dry soil). As a consequence, the sorptivity and hf are not very sensitive to the matrix potential of the dry soil. Furthermore, hf varies from a few cm to about -30 cm in clayey soils. This is of course much less negative then the matrix potentials in dry soils. As a consequence, using the matrix potential of the dry soil as hf will lead to a strong underestimation of the pressure head at the wetting front and a strong overestimation of the infiltration that is driven by capillarity. In order to compensate for this overestimation of infiltration, the saturated conductivity must be reduced to match the measured infiltration rates. This may be the reason why the skin factors are reduced when the initial soil moisture content is lower.

A: We totally agree that the reduced skinfactor values may be caused by improper use of the hydraulic conductivity and the matric potential. While the matric potential in EROSION-3D is dependent on soil moisture of dry soil, the hydraulic conductivity equals the saturated hydraulic conductivity without reduction with decreasing soil moisture. The skinfactor therefore compensates for this feature of the infiltration module of EROSION-3D. Unfortunately we are not able to address this issue by changing implemented algorithms, as EROSION-3D is a closed source tool maintained by a third party, but we will make a request with the developers to improve the implementation at this point. By a quick check of two other modelling tools for soil erosion, it seems that there are many different ways to implement the Green-Ampt infiltration approach. OpenLisem follows a comparable approach to EROSION-3D using saturated hydraulic conductivity, but allows wetting front suction as direct input parameter (https://blog.utwente.nl/lisem/basic-theory/infiltration/). SWAT uses a CN factor and saturated conductivity to estimate effective hydraulic conductivity and wetting front matric potential is a function of porosity, percent sand and percent clay (https://swat.tamu.edu/media/99192/swat2009-theory.pdf, p.108). From this we think that the outcomes of each infiltration model should be checked carefully and wherever possible should be validated. For the revised paper we will add a paragraph to the discussion section addressing this issue.

Discussion section of the paper is completely changed, this problematic is discussed, an alternative method of model parametrization is tested and future development of the model in this direction is suggested

R: P15 ln 37: 'This method reduces the number of experiments' Do you mean: the number of parameters?

A: In the previous studies, the whole dataset with experiments was split into smaller subsets based on manual decision. Regression equations to predict skinfactors were calculated based on low number of experiments in each subset.

This section was removed in accordance with other reviewer's comments

R: P 15 ln 38: 'Previous studies determined different dependencies for the prediction parameters (e.g., the intercept of soil moisture) on each single subset, whereas this study assumed an equal dependency on each parameter for the entire dataset.' But in the STEP1 model, you considered interactions between the categorical and continuous predictors.

A: You are right, we will correct it. The difference is, that in previous studies only categorical predictors were included in the analysis. Experiments were grouped into various subsets based on categories of different predictors and relationships between skinfactor and the predictors were searched within the subset. The interactions (subsets) were predefined by the authors without checking for its statistical reliability. In our approach, continuous and categorical predictors were put together, one initial complex model was applied on the whole dataset, including interactions between continuous and categorical predictors and the best model was then selected by a selection procedure. Only significant predictors and their interactions are included in the STEP1 model.

This section was removed in accordance with other reviewer's comments

R: P 16 ln 17: 'Other significant predictors of soil texture' You do not predict soil texture but you use soil texture as a predictor. 'Other significant predictors such as soil texture'

A: Yes, thank you.

Corrected

**List of other changes**

In the context of reviewer's comments, mainly the issue with the implementation and parametrization of Green-Ampt in EROSION-2D/3D model some sections are seen as less important and were either removed or abridged.

- In Skinfactor prediction section paragraph about first insight in the data was removed
- Rainfall-runoff data section was abridge (less detail about choosing final datasets, reduction of factorial variables mentioned only briefly
- Section about reproducibility of the analysis by automatized R scripts removed from introduction
- Model STEP3 was removed (it is very similar as STEP1 and STRONG)

---

## Editor Decision (ED1)

I simulated infiltration in a silty soil with Hydrus and used the following parameters of the van Genuchten model:

| θr | θs | α (1/cm) | n | Ks (cm/d) | l |
|---|---|---|---|---|---|
| 0.034 | 0.46 | 0.016 | 1.37 | 6 | 0.5 |

Infiltration was simulated in a soil profile with an initially uniform pressure heads. Initial pressure heads of -200, -500, -1000 and -5000 cm were considered. A constant head infiltration (h= 0 cm) at the surface was considered.

The simulated cumulative infiltration curves were subsequently used to fit the Green Ampt infiltration model (GA model).

$$I(t) = K_s t + (\theta_s - \theta_{ini})|h_f| \ln\left(\frac{I(t)}{(\theta_s - \theta_{ini})|h_f|} + 1\right)$$

This model was rewritten so that the time is the dependent variable and the cumulative infiltration the independent variable. Parameters were then fitted by minimizing the squared difference between the Hydrus simulated times and the times predicted by the GA model. Three different fitting scenarios were considered:

1) Both Ks and hf were fitted
2) Ks was fitted and hf was assumed to be equal to the initial pressure head
3) hf was fitted and Ks was put equal to Ks used in the Hydrus simulations

Option 2 is what I presume is implemented in EROSION 2/3D and what the authors used to analyze the infiltration curve.

What can be observed is that:

1) Option 1 and 3 fit the simulated curves better than option 2
2) When option 1 and 3 are used, the fitted hf is much smaller (in absolute value) than the initial pressure head vary only little with the initial pressure heads.
3) When option 2 is used, the fitted saturated conductivity decreases drastically with drier initial soil conditions and are much smaller that the true Ksat. This is what the authors also observe but attribute it to a decreased conductivity of a dry soil and the formation of a skin layer. The simulation results in fact show that this is incorrect interpretation of infiltration is soils.

[Figure]

*Figure 1: Infiltration curve for an initial pressure of -200 cm*

[Figure]

*Figure 2: Infiltration curve for an initial pressure of -500 cm*

[Figure]

*Figure 3: Infiltration curve for an initial pressure of -1000 cm*

[Figure]

*Figure 4: Infiltration curve for an initial pressure of -5000 cm*